# A Review: Methodologies to Promote the Differentiation of Mesenchymal Stem Cells for the Regeneration of Intervertebral Disc Cells Following Intervertebral Disc Degeneration

**DOI:** 10.3390/cells12172161

**Published:** 2023-08-28

**Authors:** Takashi Ohnishi, Kentaro Homan, Akira Fukushima, Daisuke Ukeba, Norimasa Iwasaki, Hideki Sudo

**Affiliations:** 1Department of Orthopedic Surgery, Faculty of Medicine and Graduate School of Medicine, Hokkaido University, Sapporo 060-8638, Japan; takashi.onishi.ortho@gmail.com (T.O.); kentaro.homan@gmail.com (K.H.); 621akira.fuku@gmail.com (A.F.); niwasaki38@yahoo.co.jp (N.I.); 2Department of Orthopedic Surgery, Hokkaido University Hospital, Sapporo 060-8648, Japan; daisuke922@nifty.com; 3Department of Advanced Medicine for Spine and Spinal Cord Disorders, Faculty of Medicine and Graduate School of Medicine, Hokkaido University, Sapporo 060-8638, Japan

**Keywords:** intervertebral disc degeneration, regeneration, mesenchymal stem cell, differentiation

## Abstract

Intervertebral disc (IVD) degeneration (IDD), a highly prevalent pathological condition worldwide, is widely associated with back pain. Treatments available compensate for the impaired function of the degenerated IVD but typically have incomplete resolutions because of their adverse complications. Therefore, fundamental regenerative treatments need exploration. Mesenchymal stem cell (MSC) therapy has been recognized as a mainstream research objective by the World Health Organization and was consequently studied by various research groups. Implanted MSCs exert anti-inflammatory, anti-apoptotic, and anti-pyroptotic effects and promote extracellular component production, as well as differentiation into IVD cells themselves. Hence, the ultimate goal of MSC therapy is to recover IVD cells and consequently regenerate the extracellular matrix of degenerated IVDs. Notably, in addition to MSC implantation, healthy nucleus pulposus (NP) cells (NPCs) have been implanted to regenerate NP, which is currently undergoing clinical trials. NPC-derived exosomes have been investigated for their ability to differentiate MSCs from NPC-like phenotypes. A stable and economical source of IVD cells may include allogeneic MSCs from the cell bank for differentiation into IVD cells. Therefore, multiple alternative therapeutic options should be considered if a refined protocol for the differentiation of MSCs into IVD cells is established. In this study, we comprehensively reviewed the molecules, scaffolds, and environmental factors that facilitate the differentiation of MSCs into IVD cells for regenerative therapies for IDD.

## 1. Introduction

The intervertebral disc (IVD) connects adjacent vertebrae to enable omnidirectional segment motion and absorbs a compressive load or various stains to support the spinal column [1]. Despite the benefits of utilizing this unique tissue, it exhibits complexity and delicacy. IVD degeneration (IDD) occurs secondary to genetic factors [2,3,4,5,6], mechanical overload [7,8,9,10], trauma [11,12,13,14,15], or aging [16,17] and may cause nociceptive pain in the back or neurological deficits, including neuralgia, numbness, and muscular weakness [18]. Currently available treatments include pharmacotherapies using general or neurotropic analgesics, physical therapies, and surgical treatments. These treatments are effective in alleviating symptoms but do not necessarily address the fundamental pathological conditions underlying the disease [19,20]. Furthermore, pharmacotherapies and/or physical therapies can be ineffective when the severity of IDD and accompanying neurological deficits are very advanced. Surgical treatments can often salvage these severe cases, but complications such as relapse of herniation of the nucleus pulposus (NP) [21] or adjacent segment disease after spinal fusion surgeries [22] become problematic in some cases. Accordingly, the ideal treatment for IDD is tissue regeneration, which aims to avoid or minimize the occurrence of sequential detrimental complications. As spine surgeons, we treat patients with IDD every day and encounter such complications, underscoring the importance of developing regenerative treatment for IDD.

Cell therapy has become the mainstream regenerative treatment for degenerated IVD. The World Health Organization has supported regenerative therapies that utilize mesenchymal stem cells (MSCs) and bioscaffolds as primary research objectives [23]. Implanted MSCs have been reported to exhibit anti-inflammatory, anti-apoptotic, and anti-pyroptotic effects, as well as promote extracellular component production and eventual differentiation into IVD cells [24,25,26,27,28,29,30]. While numerous NP cells (NPCs) are required in the environment for the chondrogenic differentiation of MSCs, only a small number of MSCs is sufficient to significantly enhance the proliferation of NPCs, and numerous MSCs are required for the upregulation of aggrecan expression in NPCs [29]. Collectively, the goals of MSC therapy can be recapitulated by the recovery of IVD cells and the consequent regeneration of the extracellular matrix (ECM) of the degenerated IVDs.

Focusing on the harsh environment in the NP, environmental factors, including high osmolarity and low pH, are detrimental for implanted MSCs to survive and express their biological behaviors [31]. The survival of implanted MSCs is an important aspect that may affect treatment outcomes. A previous study reported that human MSCs survived in porcine IVD for at least 6 months, as confirmed by the expression of typical chondrocyte markers [32]. However, it remains unclear whether these anabolic effects persist. Implantation of NPCs is an alternative option. Several clinical trials have previously reported that autologous NPC implantation to degenerate IVDs suppresses the progression of IDD and/or reduces disability levels of patients, in accordance with a reduction in their low back pain [33,34,35]. However, the major obstacle to using autologous NPCs is the difficulty in obtaining donor cells. Unless a plausible reason exists to harvest the NP tissue from an intact IVD, treatment of the degenerated IVD requires sacrificing another intact IVD. For this reason, previous clinical trials that had utilized degenerated IVDs to obtain donor cells and that had selected or activated those cells prior to implantation have only exhibited modest outcomes [33,34,35].

A solution to this complicated problem may be the establishment of a methodology to differentiate allogeneic MSCs in vitro and/or in vivo. MSCs may induce in vitro differentiation into healthy NPCs or annulus fibrosus (AF) cells (AFCs) that have sufficient viability, proliferative potential, and capability for ECM development in the implanted regions. Alternatively, MSC implantation can be performed using strategies aimed at differentiating cells into NPCs or AFCs in situ. This includes implantation in combination with appropriate growth factors, scaffolds, or carriers. The production of decellularized NP or AF matrices is a possible option for in vitro differentiation of MSCs in the production of bioscaffolds.

Based on this background information, we comprehensively reviewed the methodologies underlying the induction of the differentiation of MSCs into IVD cells for regenerative therapies for IDD, including strategies that employ molecules, scaffolds, and environmental factors.

## 2. Efficacies of MSCs on the Pathology of IDD

In degenerated IVD, pro-inflammatory cytokines are upregulated and trigger regulated cell death and ECM degradation [36,37,38,39]. Cell loss due to regulated cell death and phenotypic changes into hypertrophic chondrocytes leads to ECM fibrosis [19,20,40,41,42,43]. Cell loss in the NP causes a concomitant reduction in proteoglycans with a secondary reduction in hydration and hydrostatic pressure from the NP matrix [43]. Fibrocartilaginous changes in the AF matrix coincide with the loss of fiber tension, buckling, and fissure formation [44]. Cellular senescence also plays an important role in IDD [45,46,47,48]. This process induces the arrest of cellular proliferation, chronic inflammation, and ECM degradation [49]. Many inflammatory factors manifest nociceptive effects, and increased levels of nerve growth factor and brain-derived neurotrophic factor are released in degenerated IVDs [50]. In MSC therapy for IDD, many underlying conditions are affected, including anti-inflammatory, anti-apoptotic, and anti-pyroptotic effects and the promotion of ECM production, in addition to an increase in the number of IVD cells due to MSC differentiation [51,52]. A previous study reported that bone marrow-derived MSCs (BMSCs) promote an increase in endogenous notochordal cells in the NP [30]. Similarly, the conditioned medium of umbilical cord-derived MSCs (UCMSCs) was reported to recover the stemness of NP-MSCs, which is represented by an increase in CD29 and CD105 proteins with an accompanying elevation of *OCT4*, *Nanog*, and *TIE2* mRNAs [53], indicating the effects of extracellular vesicles. This process improves cellular proliferation and chondrogenic differentiation [53]. In addition to the effects of exocytosis/endocytosis, phenomena such as tunneling nanotubes reportedly contribute to subcellular component delivery from MSCs to NPCs, ultimately leading to phenotypic alteration of NPCs [54]. MSCs differentiated into either NPC- or AFC-like cells are considered to regenerate ECMs in the implanted regions. NP and AF have discrete mechanical properties due to endurance in different strains; namely, NP mainly endures compressive loads [1] and AF, tensile, or torsional strains [55]. NPCs produce the ECM enriched by type II collagen, aggrecan, and various small leucine-rich repeat proteoglycan; AFCs are the ones enriched by type I and II collagens, elastin, and fibrillin-1, and all of which characterize the properties of the forming ECM [44]. Although the majority of studies pursue MSC differentiation into NPC-like cells, several studies aimed to differentiate MSCs into AFC-like cells, which are discussed in the later section. The contents of such research studies are summarized in Figure 1.

## 3. Types of MSC Based on Its Source

Representative types of MSC, based on their source, include BMSC [56,57], adipose-derived MSC (ADMSC) [58,59], NP-derived MSC (NPMSC) [60,61], and UCMSC [53,62]. Gou et al. comparatively explained the features of each cell type in their review article [63]. Briefly, harvesting of BMSCs historically has required invasive procedures; however, evidence of improved isolation, culture, and cell therapy using these cells is accumulating [63]. ADMSCs can be obtained abundantly without highly invasive procedures and exhibit low immunogenicity [63]. NPMSCs can be stimulated to proliferate and differentiate in vitro [60] or in situ but may possess suboptimal functions when targeted to degenerated IVDs [63]. Reportedly, UCMSCs possess functions comparable to other types of MSCs [62,64]. Nevertheless, UCMSCs may have a limited chance of application in autologous implantation [63]. However, a recent study successfully accomplished a clinical trial of 1% HA-mounted allogeneic UCMSC implantation to IDD patients with low back pain [65]. Until two years after the injection, the visual analog scales of low back pain significantly reduced, whereas the index of quality of life (Oswestry Disability Index) of the patients significantly improved [65]. Limited studies have compared the functional superiority of different types of MSCs in terms of the differentiation potential into IVD cells. However, a previous study demonstrated that ADMSCs outperformed BMSCs in terms of their ability to differentiate into NPC-like cells in 3D culture, with respect to proliferation, glycosaminoglycan (GAG) and proteoglycan synthesis, and mRNA and protein expression of HIF1-α, GLUT1, SRY-Box Transcription Factor 9 (SOX9), aggrecan, and type II collagen [66]. In addition, Vadala et al. did not detect BMSCs in the NP 3 weeks after direct injection to the IVDs in a rabbit IDD model, and no sign of regeneration, except osteophyte formation, was evident [67]. Interestingly, regarding the differentiation potential of the AF-like phenotype, BMSCs were deemed superior to ADMSCs, with significantly earlier increased expression of *COL I*, *COL II*, and *ACAN* [68].

Advanced technology has enabled the sorting of highly proliferative BMSCs, such as rapidly expanding clones (RECs), based on the cell surface markers CD271 and CD90 [57]. These cells exhibit less variability and more uniform phenotype/function compared to commercial human BMSCs, thus potentially allowing improved quality for cell therapy [57].

## 4. Factors to Induce the Differentiation of MSCs into IVD Cells

To elicit the potential for MSCs to differentiate into IVD cells of interest, multiple factors can be applied, including molecules, scaffolds, and environmental factors. This section introduces each factor and organizes previously investigated findings. Figure 2 summarizes the content of this whole section.

### 4.1. Molecules—Growth Factors

Transforming growth factor (TGF)-β3 is a growth factor contained in the classical chondrogenic differentiation medium with L-proline, pyruvate, insulin–transferrin–selenium solution, and L-ascorbic acid 2 phosphate [64,69,70]. TGF-β3 is studied to differentiate MSCs into both NPCs and AFCs. A spheroid culture of BMSCs with TGF-β3, dexamethasone, and ascorbate led to the positive expression of type II collagen and *ACAN*, *DCN*, *FMOD*, and *COMP*, which was similar to levels expressed in NP tissue [71]. This culture was then used to assist the co-culturing of ADMSCs and NPCs in differentiating ADMSCs toward the NPC phenotype [72]. Other growth factors, such as bone morphogenic protein (BMP)-2 and insulin-like growth factor (IGF)-1, have been used in combination to synergistically support the effects of TGF-β3 [73,74]. The combination of TGF-β3 and BMP-2 induced chondrogenic differentiation in alginate bead-encapsulated BMSCs that had been cultured in a serum-free medium with a marked upregulation of *ACAN* and *COL2A1* [73]. IGF-1 was also synergistically affected along with TGF-β3 and enabled the differentiation of NPMSCs into NPCs, partially via the activation of the MAPK/ERK signaling pathway [74]. Gruber et al. aimed to induce the differentiation of MSCs toward AFC-like phenotypes. In combination with a 3D culture, TGF-β3 supplementation resulted in the chondrogenic differentiation of ADMSCs [75,76].

Another isoform of TGF-β, TGF-β1, is reported to contribute to MSC differentiation toward an NP/chondrogenic phenotype [59,77]. BMSCs can differentiate into NPC-like phenotypes in 3D nanofibrous poly-L-lactide scaffolds under 2% O2 hypoxia in the presence of TGF-β1 [77]. Risbud et al. also reported that hypoxia augmented the effect of TGF-β1 [78]. Hypoxia played a role in maintaining the expression of endoglin, which is the TGF-β receptor in rat MSCs that had been cultured in 3D alginate hydrogels [78]. Furthermore, TGF-β1 treatment upregulates MAPK activity, specifically ERK1/2, *SOX9*, *ACAN*, and *COL2* gene expression [78]. The synergistic effects of TGF-β1 with growth differentiation factor (GDF) 5 promoted human ADMSCs to differentiate into an NP-like phenotype, as shown by gene expression pattern and ECM production, which were determined to be via the Smad 2/3 signaling pathway [59,79]. Meanwhile, Notch 1 knockdown supported the effect of TGF-β1 regarding the enabling of the chondrogenic differentiation of MSCs [80]. Chondrogenic differentiation was also observed in TGF-β1-transfected BMSCs that had been cultured in calcium alginate gel microspheres under simulated microgravity conditions using a rotary cell culture system [81]. Some studies previously reported the efficacy of platelet-rich plasma (PRP) [82] in inducing chondrogenic differentiation or differentiating MSCs into an NP-like phenotype [83,84]. The effects of PRP are considered dependent on growth factors, including TGF-α and β, platelet-derived growth factors [85], and vascular endothelial growth factors. However, aggrecan, collagen types I and II, and SOX9 were less expressed in terms of gene and protein levels when MSCs were cultured with PRP compared to simple TGF-β1, indicating that PRP may not be recommended for MSC differentiation [84].

Other members of the TGF-β superfamily include BMPs [86], and several isoforms have been reportedly involved in the differentiation of MSCs into an NP-like phenotype or chondrogenic differentiation. BMP-2 was utilized with simulated periodic mechanical stress and a chondrogenic differentiation medium and exerted a positive effect on NPMSC differentiation toward NPC [70]. Utilizing BMP-2 in combination with TGF-β3 was found to adequately enhance the chondrogenic differentiation of BMSCs that had been cultivated in alginate beads in a serum-free medium [73]. BMP-2-transduced BMSCs cultured in PRP gels also promoted the chondrogenic differentiation of BMSCs [87]. Another isoform, BMP-3 supplementation after pretreatment with IL-1β, was proven to enhance human MSC proliferation and chondrogenic differentiation [88]. BMP-7 was overexpressed in BMSCs via vector transduction, which induced NP-like differentiation through the Smad pathway [89]. In a comparative study of BMP-2 and BMP-7, BMP-2 was suggested to induce osteogenic differentiation, whereas BMP-7 induced chondrogenic differentiation of ADMSCs [90]. *RUNX2* and *SPP1* were found to be upregulated by BMP-2 but not BMP-7, and *ACAN* was found to be upregulated only by BMP-7 treatment [90], suggesting that BMP-2 induces osteogenic rather than chondrogenic differentiation.

GDF5 and 6 are expected to replace TGF-β regarding the efficacy of differentiating MSCs into an NPC-like phenotype. GDF5 transfection or supplementation exhibited an effect on NPMSC [91] and BMSC [79,92,93] differentiation into NP-like cells. In another study, GDF5 was electroporated into BMSC cultured in 1.2% alginate beads, which successfully exhibited chondrogenic differentiation [94]. Human recombinant GDF6 has been reported to differentiate both BMSCs and ADMSCs into an NP-like phenotype [95]. Notably, a previous study showed that GDF6 outperformed GDF5 or TGF-β3 in terms of the differentiation potential of both BMSCs and ADMSCs into an NP-like phenotype [96]. Other studies have investigated the effect of GDF6 on MSCs embedded in carriers, such as poly(DL-lactic acid-co-glycolic acid), (PLGA)-polyethylene glycol-PLGA microparticles, or a combination of poly(N-isopropylacrylamide-graft-chondroitin sulfate) hydrogel and alginate microparticles, and revealed the role of BMSC and ADMSC differentiation toward an NP-like phenotype [97,98].

Several other growth factors have also been reported to play a role in MSC differentiation. Insulin-like growth factor (IGF)-1 supplemented the culture of human MSCs to render NPC-like differentiation [85,99]. Fibroblast growth factor (FGF)-2 is a potent mitogenic factor and, when cultivated in alginate, is reported to play a role in maintaining the NPC phenotype via a TGF-β1 response [100]. It also induces MSC differentiation into either the NPC-like or chondrogenic phenotypes [85,101]. However, the opposite effect of FGF-2 has also been reported; namely, novel NP markers decreased in FGF-2-supplemented cultures. This suggests that FGF-2 exhibits an overall controversial role in terms of MSC differentiation into an NPC-like phenotype [102]. Table 1 summarizes the content of this section.

### 4.2. Molecules—Other Endogenous Factors

Wnt, a cysteine-rich endogenous glycoprotein, is encoded by 19 genes of the human genome [103,104]. Wnt signaling is recognized as an important player during IVD development and has pivotal effects depending on canonical or noncanonical signaling as well as cell- and tissue-specific signaling [105]. Focusing on the chondrogenic differentiation of MSCs, the effect of Wnt3a is controversial [105]. Although various growth factors, such as TGF-β1, 3, BMP-2, and FGF-2, are used in combination, some studies have shown positivity [106,107] and others have shown negativity [108,109,110,111]. In contrast, Wnt5a positively affected the chondrogenic differentiation of MSCs [105,106,111,112,113]. Treatment with lithium chloride promotes the differentiation of ADMSCs toward the NPC-like phenotype, presumably due to augmentation of the glycogen synthase kinase 3β-dependent β-catenin/Wnt pathway [114].

Silent mating type information regulator 2 homolog 1 (SIRT1) is an NAD^+^-dependent deacetylase that deacetylates histones and other molecules [115,116]. It is involved in a broad range of cellular processes such as apoptosis, autophagy, and inflammation; however, its role in preventing cell senescence and prolonging the lifespan of an organism is especially underscored [115,116,117]. SIRT1 promotes the chondrogenic differentiation of NPMSCs by downregulating the monocyte chemoattractant protein 1 and chemokine receptor 2 axis [117].

SOX9 regulates MSC differentiation into chondrocyte-like cells [118]. The conditional knockout of *SOX9* in *ACAN*-expressing cells resulted in the progressive degeneration of all compartments of the IVD, including the cartilaginous endplate [119]. When *SOX9* was transfected into BMSCs cultivated in porous biodegradable three-dimensional (3D) poly-L-lactic acid scaffolds, the cells differentiated into an NPC-like phenotype, generating type II collagen and aggrecan [118]. Sine oculis homeobox homolog 1 (SIX-1) is a transcription factor that is expressed during the development of limb tendons [120,121]. In a study in which *SOX9* and *SIX1* were overexpressed in UCMSCs, the cells ultimately exhibited chondrogenic differentiation with an enhancement in the expression of *TGFB1*, *BMP*, *SOX9*, *SIX1*, and *ACAN* [122].

Mohawk (Mkx) is a homeobox protein that is a key transcription factor and regulator of AF development, maintenance, and regeneration, and is mainly expressed in the outer AF [123]. Accordingly, *Mkx* was overexpressed in MSCs to ultimately determine whether differentiation into AFC-like cells occurred. The results indicated that MSCs were differentiated toward the AFC-like phenotype, thereby resulting in enhanced type I collagen and decorin mRNA and protein levels, possibly via the TGFβ/Smad signaling pathway rather than the BMP/Smad signaling pathway [123].

Coenzyme Q10 (Co-Q10) is an endogenous lipophilic molecule, and also known as ubiquinone (2,3-dimethoxy-5-methyl-6-polyprenyl-1,4-benzoquinone) [124]. It is found in the phospholipid bilayer of cellular membranes and is especially localized in the mitochondrial inner membrane, where it serves as a component of the mitochondrial electron transport chain [124,125]. The main effect of Co-Q10 is the inhibition of mitochondrial ROS generation and the subsequent prevention of cellular senescence, which is also applicable to stem cells [126]. To resolve the challenges of utilization due to the hydrophobic nature of Co-Q10, it was coated with a phospholipid molecule, namely lecithin, to render it hydrophilic and treat BMSCs. The results showed that Co-Q10 protected BMSCs from oxidative stress and promoted their differentiation toward an NP-like phenotype [124].

Link N is the N-terminal peptide of the link protein that stabilizes the interaction between aggrecan and hyaluronan [127]. It is generated during proteolytic degeneration in vivo and has an agonistic effect on collagen synthesis in NP and AF pellet cultures [127,128]. Although Link N alone did not induce MSC chondrogenesis, it was inductive when applied together with a chondrogenic differentiation medium, resulting in increased GAG secretion, the upregulation of *ACAN*, *COL2A1*, and *SOX9* expression, and the downregulation of *COL10A1* and *BGLAP* expression [129].

MicroRNAs (miRNAs) are small non-coding RNAs comprising 15–30 nucleotides that function as post-transcriptional inhibitors of gene expression. Numerous studies have reported that they play an important role in the process of IDD [43,130]. Most miRNAs are involved in the promotion or suppression of regulated cell death in IVD cells [43]; however, a few contribute directly to the differentiation of MSCs. miR-15a is also known to modulate the expression of genes involved in cellular proliferation and apoptosis [43,130]. Moreover, this miRNA has been studied for its role in the chondrogenic differentiation of NPMSCs. It was transfected into NPC-derived exosomes and used to treat NPMSCs, resulting in increased aggrecan and type II collagen mRNA and protein levels, whereas mRNA and protein levels of ADAMTS4/5 and MMP-3/-13 decreased [131]. Further studies revealed that this effect is mediated through the PI3K/Akt and Wnt3a/β-catenin axes [131]. Another miRNA, termed miR-140-3p, is also downregulated in degenerative IVDs [132,133]. Based on this study, the effect of miR-140-3p overexpression on the progression of IDD was assessed. The overexpression of miR-140-3p alleviates IDD by targeting Kruppel-like factor 5 (KLF5), which interferes with the migration and differentiation of MSCs [133]. NPMSCs from degenerated IVDs were facilitated to differentiate into NPCs through the inhibition of the KLF5/N-cadherin/mouse double minute 2/Slug axis [133].

The role of serum supplementation in cell culture remains largely unknown, although numerous humoral factors are thought to be involved [134]. Interestingly, serum deprivation seemed to be optimal for inducing chondrogenic differentiation of MSCs. The conditions for culturing ADMSCs with or without fetal bovine serum (FBS) were evaluated [135]. Although FBS-free conditions allow ADMSCs to survive, proliferate, and undergo adipogenic, osteogenic, and chondrogenic differentiation, ADMSCs cultured without FBS have enhanced potential for chondrogenic differentiation [135]. Table 2 summarizes the content of this section.

### 4.3. Molecules—Exogenous Factors

Ortho-vanillin (o-vanillin) is a natural compound that inhibits toll-like receptors, thereby preventing inflammation [136]. O-vanillin exhibits senolytic properties and augments the proliferation of non-senescent cells, which consequently increases ECM synthesis in degenerated IVDs [137]. In another study, the conditioned medium of o-vanillin-treated human IVD cells (NPCs and inner AFCs) induced the chondrogenic differentiation of human MSCs, as shown by the elevation of *FOXF1*, *PAX1*, *TIE2*, *SOX9*, *HIF1A*, and *ACAN* gene expression compared to the control [138].

BuShenHuoXueFang (BSHXF) is a Chinese herbal formula that has been reported to improve the environment of degenerated IVD, enhance NPC proliferation, and delay IDD progression [139]. Therefore, the role of BSHXF-medicated serum in MSC differentiation was examined, and ADMSCs exhibited differentiation toward an NPC-like phenotype [140].

Asperosaponin VI (ASA VI) is an herbal Chinese traditional medicine with a long history of safe use in strengthening tendons and bones [141]. The ERK1/2 and Smad2/3 signaling pathways regulate the differentiation of NPMSCs into NP-like cells [74], and ASA VI modulates these pathways [142]. Hence, ASA VI was assessed for its effects on human MSCs, and it was confirmed that MSCs differentiate into NP-like cells [141].

Salvianolic acid B is a compound of *Radix Salvia miltiorrhiza* extracted from the roots of *S. miltiorrhiza* and is similar to “Danshen”, which is another traditional Chinese medicine [143]. It is known as a reactive oxygen species scavenger and an inhibitor of inflammation and metalloproteinase expression in aortic smooth muscle cells [143]; therefore, it has been used to treat cardiovascular diseases in China [144]. Based on previous studies, salvianolic acid B was assessed whether it promotes MSC differentiation in the context of NP regeneration. Salvianolic acid B treatment increased the type II collagen, proteoglycan, TGF-β1, and water content of MSC-implanted IVDs compared to the control, suggesting its ability to enhance the chondrogenic differentiation of MSCs in vivo [145].

Psoralidin (PSO) is the main bioactive compound in the traditional medicine, *Cullen corylifolium* (L.) Medik [146]. PSO has been identified in the seeds of medicinal plants. Cullen corylifolium grows in Asia, India, and Europe [147]. PSO has various anti-inflammatory, antibacterial, antioxidant, antipsoriatic, antidepressant, estrogenic-like, and antitumor properties, and may also stimulate osteoblast proliferation [146]. Considering the results of previous studies, PSO was investigated for its effect on ADMSC differentiation, and differentiation toward an NPC-like phenotype was confirmed [148].

Simvastatin is an approved medicine for hyperlipidemia; however, previous studies have elucidated its effect on inhibiting NPC apoptosis and preventing IDD [149,150]. Furthermore, simvastatin was reported to drive osteogenic differentiation and the migration of BMSCs [151,152]. Based on these results, the effect of simvastatin was explored on the differentiation potential of NPMSCs. This research demonstrated that NPMSCs successfully differentiated into NPC-like phenotypes following treatment with simvastatin [61].

Pentosan polysulfate (PPS) is a semi-synthetic sulfated xylan isolated from beech trees that acts similarly to heparan sulfate in vivo [153]. It has been used to treat interstitial cystitis [154] and is an anti-arthritic drug for coxalgia [153]. The mechanism of this anti-inflammatory effect is considered to be the inhibition of complement activation via C-reactive proteins and the aggregation of IgG [155]. In addition, PPS regulates coagulation [156], fibrinolysis [157], thrombocytopenia [158], the synthesis of hyaluronan [159], the inhibition of nerve growth factor production in osteocytes [160], and the stimulation of proteoglycan synthesis in chondrocytes [161,162]. This multifactorial mucopolysaccharide derivative also has the potential to induce the chondrogenic differentiation of BMSCs. After treating BMSCs with PPS, PPS was successfully internalized by BMSCs and consequently augmented both cell proliferation and proteoglycan synthesis [163,164]. The application of PPS-treated BMSCs to degenerated IVDs with a collagen sponge inhibited the IDD processes in an ovine model of lumbar microdiscectomy [164]. Table 3 summarizes the content of this section.

### 4.4. Cellular Engineering

The surface of ADMSCs was functionalized with unnatural sialic acid via metabolic glycoengineering, and it was examined whether this cellular engineering improved the specificity of ADMSC differentiation toward the NPC-like phenotype. The results showed elevated NPC markers, namely *SOX9*, *COL2*, *KRT19*, and *CD24* expression [165]. Consistently, the implantation of glycoengineered ADSCs improved the height, biomechanical properties, and histological findings of the treated IVDs [165].

### 4.5. Conditioned Mediums, Exosomes, and Co-Cultures

Interactions between different cell types cause reciprocal phenotypic changes. Humoral factors secreted from cells, either directly or indirectly encapsulated in extracellular vesicles, such as exosomes, are a form of cellular communication [166,167]. Another method of cellular communication involves tunneling nanotubes, in which the transfer of subcellular materials occurs [54]. In this section, we introduce methodologies for the chondrogenic differentiation of MSCs using a conditioned medium, exosome, and co-culture with IVD cells.

The conditioned medium of notochordal cells (NCCM) exhibited a strong effect on the chondrogenic differentiation of BMSCs. NCCM resulted in significantly higher GAG accumulation than either the control medium or the chondrogenic differentiation medium [168]. While the NPC-conditioned medium (NPC-CM) does not exhibit a consistent trend of MSC differentiation under normoxia, NPC-CM in combination with hypoxia (2% O_2_) consistently revealed an upregulation of *ACAN*, *TBXT*, *COL2*, *KRT8*, *KRT19*, and *SHH* in BMSCs [169].

NPC exosomes may play a factor in the effect of NPC-CM and actually promote the differentiation of BMSCs into an NPC-like phenotype, as demonstrated by the upregulation of *ACAN*, *SOX9*, *COL2A1*, *HIF1A*, *CA12*, and *KRT19* expression [170]. Moreover, the upregulation of aggrecan, type II collagen, Sox-9, CA12, and KRT19 protein levels has also been established [170]. However, the direct treatment of BMSCs with NPC exosomes was more effective in differentiating BMSCs into NPC-like cells compared to the trans-well co-culture of BMSCs with NPCs [171]. The effect of the trans-well co-culture was demonstrated to occur through NPC exosomes by confirming the role of Rab27a, an important protein in the process of exosome secretion [171]. However, the discrepancy in the potential between co-cultures and exosomes was possibly due to the lower concentration of exosomes released by NPCs in the co-culture method [171]. The effect of NPC exosomes on BMSC differentiation was further shown to be mediated by the Notch 1 pathway [171]. Although the Hypoxia/HIF-1α-Notch signaling pathway plays an important role in cell proliferation and the self-renewal of NPCs [172,173,174,175,176], the Notch signaling pathway exhibited a negative role in the expression of ECM component genes, including *COL2*, *ACAN*, and *SOX9* [171]. Similar results were confirmed by Notch1 knockdown in combination with TGF-β1 treatment, resulting in the upregulation of proteoglycan and type II collagen expression in rabbit MSCs [80].

The co-culture of MSCs with NPCs is often used to differentiate MSCs into NPC-like cells. Both direct co-culture [56,57,177,178,179,180] and trans-well co-culture [72,181,182,183] successfully induced the chondrogenic differentiation of MSCs. Wharton’s jelly is another source of MSCs [184]. Both the direct and trans-well co-cultures of Wharton’s jelly cells with NPCs induced the differentiation of MSCs to NP-like cells, but the gene expression levels of *ACAN*, *COL2*, and *SOX9* were higher in the direct co-culture group [184]. The co-culture of these cells in special settings has also been investigated. Synergistic effects on chondrogenic differentiation were observed in the dynamic compression and co-culture of ADMSCs with NPCs at a 12 h intermittent dynamic hydrostatic pressure of 17 kPa [185]. In the bilaminar cell pellet, where a sphere of MSCs forms the core and shell, NPCs increased MSC proliferation and chondrogenic differentiation compared to single cell-type pellets or randomly mixed co-culture pellets [186]. The co-culture of MSCs with AFCs has also been confirmed to induce MSC differentiation toward AFC-like cells. Similarly, both direct co-culture [76] and trans-well co-culture [75] successfully induced MSC differentiation into AFC-like cells. A comparison of differentiation efficiency toward AFC-like phenotypes revealed the superiority of BMSCs over ADMSCs when direct co-culture was performed with AFCs [68]. The co-culture of rat BMSCs with IVD tissue, including the inner NP, outer AF, and part of the endplate, promoted the chondrogenic differentiation of BMSCs, as evidenced by the expression of type II collagen, aggrecan, Sox-9 mRNA, and protein levels [187]. Table 4 summarizes the content of this section.

### 4.6. Biomaterials—Scaffolds and Carriers

Numerous studies have reported the efficacy of biomaterials that function as scaffolds or carriers for implanted MSCs to undergo differentiation and/or proliferation. Hydrogels are the most widely studied materials for this purpose. Cell and hydrogel interactions influence cell reactions such as differentiation, proliferation, and migration [188]. The compositions of hydrogels are more than 90% water [188] with the following diverse additives: natural materials, including collagen [189,190], gelatin [191,192], hyaluronic acid (HA) [193], alginate [194], fibrin [195], chitosan [196,197,198], agarose [199], polypeptide [32,200], PRP [193], PRP/HA/batroxobin (anticoagulant and gelling agent reactive to PRP) [201], and multiple materials combined [202,203], and synthetic materials, including polyethylene glycol (PEG) [204], polyacrylamide [205], redox-polymerized carboxymethylcellulose [206], methacrylated carboxymethylcellulose [207], poly(N-isopropylacrylamide- N,N0-dimethylacrylamide-Laponite [208], poly(acrylamide-co-acrylic acid) microhydrogels [209], poly lactide-co-glycolide [210,211], and poly glycerol monomethacrylate-poly 2-hydroxypropyl methacrylate diblock copolymer [58]. Various hybrid hydrogels have also been studied, such as PPS incorporated PEG and HA [212], a highly sulfated semi-synthetic polysaccharide combined with PEG/HA [213], poly(N-isopropylacrylamide-graft-chondroitin sulfate) hydrogels combined with or without alginate microparticles [98,214], poly D,L-lactide-co-glycolide nanoparticles carrying TGF-β3 in dextran/gelatin hydrogels [215], 1-ethyl-3(3-dimethyl aminopropyl) carbodiimide and N-hydroxysuccinimide cross-linked type II collagen/HA hydrogels [203], and nitrogen-doped plasma-polymerized ethylenes [216].

Commercial gel matrices include fiber-forming peptide gels, Hydromatrix, and Puramatrix. They were compared in terms of their potential to enable the chondrogenic differentiation of MSCs, and Hydromatrix ultimately revealed the strongest potential [217].

Growth factors, such as BMP-2, TGF-β3, GDF-5, GDF-6, and basic FGF, are combined with various materials for release [73,94,97,98,202,210,215]. Similarly, NC-CM or NP extracts with humoral factors [192,218] and chondrogenic differentiation media [219] are used in combination with various materials.

Other types of gels (non-hydrogels or undefined) that comprise natural materials—including alginate gels [56,57,220,221], poly L-lactic acid scaffolds [118], and collagen-based carriers [219,222,223]—as well as synthetic materials—including PEG diacrylate microcryogels [224], a biocompatible KLD-12 polypeptide (ACN-KLDLKLDLKLDL-CNH2)/TGF-β1 nanofiber gel [225], and layered double hydroxide nanoparticles [226]—are also all reported to be effective in the chondrogenic differentiation process of MSCs.

A comparison among the four matrices revealed that collagen, gelatin, alginate, and chitosan, in this exact order, exhibited strong potential for MSC differentiation into an NPC-like phenotype (alginate and chitosan exhibit similar potential) [227]. Another study compared alginate and chitosan hydrogels and found that alginate generated more GAGs and type II collagen [228].

Decellularized ECM—including simple decellularized NP-ECM [229,230], genipin-cross-linked decellularized NP hydrogels [231], genipin-cross-linked decellularized AF hydrogels [232], and decellularized NP and AF ECM mixtures [233]—have been demonstrated as effective scaffolds for MSCs.

The utilization of a pellet culture may allow for a similar approach. Compared to alginate beads, the pellet culture of MSCs results in higher chondrogenic differentiation [84].

A completely chimeric material, such as a silk-based scaffold, can also enhance the NP-like or AF-like differentiation of MSCs [234,235]. The majority of studies about biomaterials are observational studies confirming the MSC differentiation into NP-like/chondrogenic phenotypes by phenotypic markers or proteoglycan staining. One study utilized energy-dispersive X-ray analysis for the confirmation of proteoglycans [208]. Some studies included biomechanical analysis; however, such results are out of the scope of the present study. Briefly, biomechanical analyses include comparing the gelatin colloidal gel of various concentrations with NP tissue to find the appropriate concentration [191], a comparison of PRP and HA/PRP at different temperatures [193], a comparison of chitosan–poly(hydroxybutyrate-co-valerate) mixed with various ratios [197], and a stiffness analysis of various concentrations of agarose gel [199] or PEGs with different molecular weights [204]. None of the material that impeded the cellular viability, e.g., the high viability of MSCs seeded in chitosan hydrogel, was confirmed [196]. Table 5 summarizes the content of this section.

### 4.7. Environmental Factors

Environmental factors, including oxygen tension, osmolarity, and mechanical stress, affect MSC differentiation.

Hypoxia is a major factor determining the fate of NPCs. Since the IVD is the largest avascular tissue in vertebrates, the NP located at the center is hypoxic [236]. Generally, hypoxia confers the transcription factor hypoxia-inducible factor (HIF)-1α to function by inactivating prolyl hydroxylase and factor-inhibiting HIF, which degrades HIF-1α via proteosomal degradation [237]. The NP is a unique tissue involved in the reaction of HIF-1α partially to oxygen tension, as PHDs play a limited role in HIF-1α degradation, thereby stabilizing HIF-1α [236,238]. Nonetheless, hypoxia and HIF-1α play essential roles in the NP and in maintaining the homeostasis of tissue-controlled metabolism [239], tissue pH [240], and the NPC-like phenotype [241]. Numerous studies have reported that hypoxia promotes MSC differentiation toward an NPC-like phenotype [77,78,79,169,242,243,244]. NPC-CM treatment exerts agonistic effects on BMSC differentiation toward NPC-like cells only when cultured in hypoxia [169]. In addition, the overexpression of HIF-1α directs ADMSCs to differentiate into an NPC-like phenotype [183].

Hyperosmolarity is a characteristic feature of the NP, along with hypoxia. Negatively charged sulfated GAGs on aggrecan attract sodium ions to the NP, leading to hyperosmolarity [245]. A key molecule in this tissue, tonicity-responsive enhancer-binding protein (TonEBP/nuclear factor of activated T-cells 5 (NFAT5)), is a transcription factor recruited in a hyperosmotic environment. It suppresses the influx of sodium by regulating the intracellular levels of nonionic osmolytes such as taurine, sodium/myoinositol, and betaine, and by modulating their transporters or synthetic enzymes [245]. TonEBP also targets cyclooxygenase (COX)-2, which is well-known as an essential enzyme in prostaglandin (PG) synthesis [246]. COX-2 also plays a role in cell survival and osmoadaptation [245]. TonEBP is a positive regulator of chondroitin sulfate and aggrecan [247,248,249]. Another target of TonEBP, heat shock protein (Hsp)70, is upregulated in the NP for cell survival in harsh environments [250]. Despite the physiological hyperosmolarity of the NP, whether high osmotic pressure promotes the differentiation of MSCs into NPC-like cells remains controversial. Some studies have demonstrated positive involvement [251,252], as 400 mOsm/L pressure simulating moderate IDD exhibited an anabolic effect, and 500 mOsm/L pressure simulating healthy IVD to suppress the NP-like differentiation of ADSCs [251,253,254]. Moreover, simultaneous hypertrophy during chondrogenic differentiation of MSCs was found to depend on the type of osmolyte used [255]. However, other studies have clearly rejected any positive involvement of hyperosmolarity (pressures within the range of 400–600 mOsm/L) compared to lower osmolarity (300 mOsm/L) in NPMSCs [253] and ADMSCs [254]. Collectively, the application of hyperosmolarity to MSC differentiation toward NPC-like cells requires additional investigation.

Mechanical stresses, such as compressive strain, are loaded into IVD in daily life [7]. In a caprine organ culture model, cell viability, cell density, and gene expression were preserved with either a low dynamic compressive load (0.1–0.2 MPa, 1 Hz) or a simulated-physiological compressive load (0.1 to 0.6 MPa, a sinusoidal load with gradual change, and on/off) [7]. NPMSCs cultured in a chondrogenic differentiation medium with BMP-2 were subjected to periodic mechanical stress (0–200 kPa, 0.1 Hz), which synergistically promoted chondrogenic differentiation [70]. Multiple studies have reported similar results, including static compression [224] and cyclic and dynamic compression [256,257,258,259,260,261,262,263]. However, excessive compressive loads as high as 1.0 MPa (static) ultimately inhibit NPMSC differentiation [264]. Table 6 summarizes the content of this section.

## 5. Discussion—Highlights, Limitations, and Future Perspectives

IDD features cell loss and ECM alteration; therefore, MSC therapy is a promising strategy to regenerate cells and tissue. Numerous studies have explored novel and efficient methodologies that enable MSC differentiation in the regeneration of IVD cells.

Among the multiple types of growth factors, TGF-β superfamily members and GDF5 and 6 played the primary roles, with BMP-7 and GDF6 likely being the most effective in eliciting MSC differentiation in NPC-like cells. However, the efficacies of other endogenous and exogenous molecular factors have not been studied. Moreover, their interactions remain unknown; for example, it is not known whether these molecules synergistically enhance each other’s effects, negate them, or have completely unrelated effects. This scarcity of information is likely the limitation of their clinical use. In contrast, biomaterials and environmental factors have been extensively studied in combination with these molecules, with most studies elucidating their synergistic effects. Hence, the combination of molecules, biomaterials, and environmental factors may be helpful in identifying a novel methodology for MSC differentiation.

Numerous types of biomaterials—including natural, synthetic, and chimeric materials—have been studied; these materials simulate scaffolds of the ECM of IVD. Although synthetic and chimeric materials can be structurally sturdy, their application in clinical use is likely limited due to the likelihood of foreign body reactions [265].

Humoral factors from notochordal cells or NPCs have been studied for their role in MSC differentiation. Interestingly, the direct treatment of BMSCs with NPC exosomes was more effective than the trans-well co-culture of BMSCs with NPCs, even though the effects of the co-culture occur via exosomes. This result indicated that purified exosomes with high yield are more inductive of MSC differentiation than the generally used co-culture, suggesting that the purified exosomes are more efficient.

An important issue to address when implanting MSCs into the NP is the uniquely harsh environment of the degenerated NP. As discussed in Section 4.7 of the manuscript, the environment of the NP has low oxygen tension, acidity, relatively high osmolarity, and detrimental mechanical stress [266]. Although hypoxia improves the chondrogenic differentiation of MSCs, other factors can impede the viability of implanted MSCs. Strategies to precondition MSCs before implantation can promote cell survival. The overexpression of HIF-1α in MSCs may promote the function of monocarboxylate transporter 4 to efflux lactate from the cells or enhance the function of carbonic anhydrase 9 and 12 to recycle bicarbonate to reside in acidity [240,267]. Mechanical overload induces apoptotic cell death, which suggests that anti-apoptotic preconditioning may help MSCs to strive against such stress, including the overexpression of *BCL2* or knocking down *CASP3* [19,268].

Major risks associated with implanted cells include potential risks of tumorigenicity, immune rejection, and long-term viability of implanted cells. However, MSCs are advantageous regarding the first two concerns, as tumorigenesis has not been reported yet. Further, the privilege of IVDs to the immune system [269,270] is well-known, and no specific serological reactions were detected in a clinical trial [271], which alleviates the concern of immune rejection. MSCs reportedly can survive in porcine IVD for at least six months [32], but cellular viability thereafter is still unclear. Further study is needed to clarify this aspect, and some successive therapy may be required to maintain viable cells.

It may be inevitable for studies to compare autologous versus allogeneic MSCs. One of these two options is likely selected based on different factors, such as invasiveness, cost, and time requirements. Autologous cells are associated with fewer concerns of immunogenicity, but they are invasive to patients, especially while harvesting patient-specific cells. Moreover, they have a high cost and require a long time to increase the cell number while maintaining good clinical practice standards. In contrast, allogeneic cells can be purchased from companies in a ready-to-use form, require no invasive procedure, and have lower costs [65,272]. In addition, multiple studies have reported that no immune response occurs with allogeneic MSC implantation [65,271], which can facilitate the clinical use of allogeneic MSCs.

As a future perspective, a multifactorial perspective may be utilized to establish more advanced strategies for differentiating MSCs into NPC- or AFC-like cells. To improve the current methodologies, combining molecules, biomaterials, and environmental factors may be a promising strategy. The concentrations or molecular weights of the materials can be determined considering the importance of cellular viability and biomechanical properties. As information and discussion about the cost-effectiveness of materials are scarce, it is important to consider this when overlooking the course to clinical application. Future studies involving biomaterials should consider the cost-effectiveness of these materials. Similarly, in the stage of translational research, studies should meet the standard of good laboratory practice and undergo a thorough and general investigation to confirm the biocompatibility and safety of the materials.

Furthermore, future studies can compare multiple types of combinations to discover the best option to advance regenerative medicine for IDD. Further, a study combining these approaches should provide more advanced methodologies for IVD regeneration.

## Figures and Tables

**Figure 1 cells-12-02161-f001:**
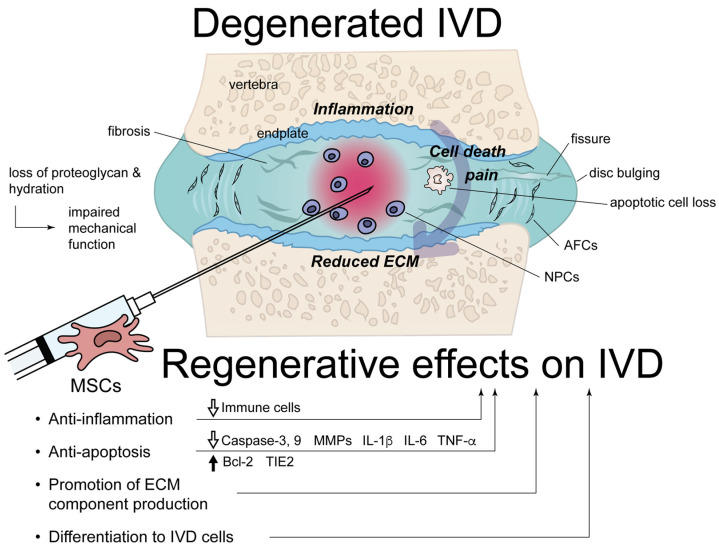
A scheme summarizing the pathological conditions of intervertebral disc (IVD) degeneration and the regenerative effects of mesenchymal stem cells (MSCs). AFC, annulus fibrosus cell; Bcl-2, B-cell lymphoma 2; ECM, extracellular matrix; IL, interleukin; MMP, matrix metalloproteinase; NPC, nucleus pulposus cell; TIE2, tyrosine kinase with Ig and EGF homology domains-2; TNF-α, tumor necrosis factor-α.

**Figure 2 cells-12-02161-f002:**
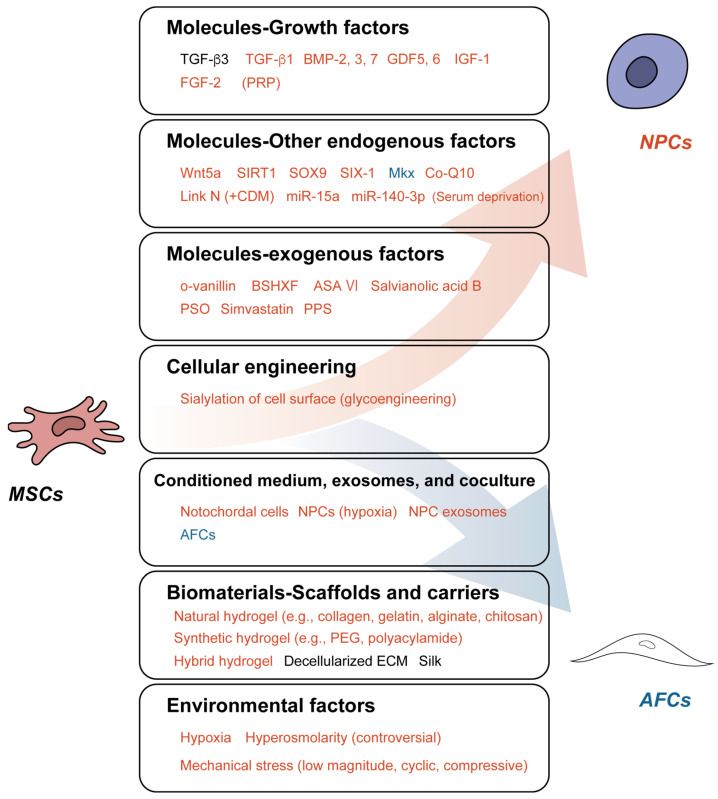
Various strategies to differentiate mesenchymal stem cells (MSCs) into nucleus pulposus cells (NPCs) and annulus fibrosus cells (AFCs). Factors in red induce MSC differentiation toward NPC/chondrogenic phenotype; factors in blue, AFC phenotype; factors in black, either NPC/AFC phenotype. ASA VI, Asperosaponin VI; BMP, bone morphogenic protein; BSHXF, BuShenHuoXueFang; CDM, chondrogenic differentiation medium; Co-Q10, Coenzyme Q10; ECM, extracellular matrix; FGF-2, fibroblast growth factor-2; GDF, growth differentiation factor; IGF-1, insulin-like growth factor-1; miR, microRNA; Mkx, Mohawk; O-vanillin, Ortho-vanillin; PEG, polyethylene glycol; PPS, pentosan polysulfate; PRP, platelet-rich plasma; PSO, psoralidin; SIRT1, silent mating type information regulator 2 homolog 1; SIX-1, sine oculis homeobox homolog-1; SOX9, SRY-Box Transcription Factor 9; TGF, transforming growth factor; Wnt5a, Wingless signaling transduction 5a.

**Table 1 cells-12-02161-t001:** Growth factors for inducing mesenchymal stem cell differentiation into intervertebral disc cells.

Growth Factor	Effects and Examples of Usage	References
TGF-β3 (10 ng/mL [69,73,74,75]; 10 μg/mL [71])	A component of the CDM; MSC differentiation to NPC and AFC in combination with BMP-2 and IGF-1	[64,69,70,71,72,73,74,75,76]
TGF-β1 (1 ng/mL [62]; 10 ng/mL [66,78]; 20 ng/mL [77])	MSC differentiation toward the NP/chondrogenic phenotype; synergistic effect with hypoxia or GDF5; the upregulation of ERK1/2 activity; the effect is augmented with Notch 1 KD	[59,77,78,79,80,81]
BMP-2 (200 ng/mL [70]; 100 ng/mL [73]; 10 ng/mL [90])	MSC differentiation toward NPC in combination with TGF-β3, CDM, and alginate beads in a serum-free medium in combination with PRP gel	[70,73,87,90]
BMP-3 (10 ng/mL)	MSC proliferation and chondrogenic differentiation in combination with pretreatment with IL-1β	[88]
BMP-7 (100–300 ng/mL [89]; 10 ng/mL [90])	MSC differentiation toward NP-like cells via the Smad pathway; better chondrogenic differentiation potential than BMP-2	[89,90]
GDF5 (100 ng/mL [92])	MSC differentiation toward NP-like cells in combination with alginate beads	[79,91,92,93,94]
GDF6 (100 ng/mL [95,96,97,98])	MSC differentiation toward NP-like cells is better than GDF5 or TGF-β3 in combination with synthetic biomaterials	[95,96,97,98]
IGF-1 (500 ng/mL [62]; 10 ng/mL [74] 100 ng/mL [99])	MSC differentiation toward NP-like cells	[62,74,85,99]
FGF-2 (10 ng/mL [100,101])	MSC differentiation to NPC-like or chondrogenic phenotypes	[85,100,101]
PRP (platelet concentration > 1 × 10^6^/μL [82])	MSC differentiation to NPC-like or chondrogenic phenotypes; contains TGF-α and β, platelet-derived growth factors, vascular endothelial growth factor, endothelial growth factor; inferior effect compared to simple TGF-β1	[82,83,84]

AFC, annulus fibrosus cell; BMP, bone morphogenic protein; CDM, chondrogenic differentiation medium; ERK, extracellular signal-regulated kinase; FGF, fibroblast growth factor; GDF, growth differentiation factor; IGF, insulin-like growth factor; IL, interleukin; KD, knockdown; MSC, mesenchymal stem cell; NP, nucleus pulposus; NPC, nucleus pulposus cell; PRP, platelet-rich plasma; TGF, transforming growth factor.

**Table 2 cells-12-02161-t002:** Endogenous factors for inducing mesenchymal stem cell differentiation into intervertebral disc cells.

Factors	Effects and Examples of Usage	References
Wnt3a (mouse [107]; lentiviral vector [109]; transfected L929 cells [110]; 5–40 ng/mL [108])	Controversial effects on the chondrogenic differentiation of MSCs	[105,106,107,108,109,110,111]
Wnt5a (retroviral vector [113])	Positively affected the chondrogenic differentiation of MSCs	[105,106,111,112,113]
SIRT1 (lentiviral vector)	It promotes the chondrogenic differentiation of NPMSCs by downregulating the monocyte chemoattractant protein 1 and chemokine receptor 2 axis	[117]
SOX9 (Adenoviral vector [118]; non-specified vector [122])	SOX9 transfected into BMSCs and cultivated in poly-L-lactic acid scaffolds resulted in BMSC differentiation into an NPC-like phenotype; use in combination with SIX-1 alternatively	[118,122]
Mkx (Retroviral vector)	Its overexpression resulted in MSC differentiation toward the AFC-like phenotype, possibly via the TGFβ/Smad signaling pathway	[123]
Co-Q10 (Bidepharm, 97% purification)	Hydrophobic lecithin-coated Co-Q10 protected BMSCs from oxidative stress and promoted their differentiation toward an NP-like phenotype	[124]
Link N (0.1 μg/mL or 1.0 μg/mL)	Link N alone did not induce MSC chondrogenesis in combination with CDM-induced MSC chondrogenesis	[129]
miR-15a (100 nmol/L, GenePharma)	Transfected into NPC-derived exosomes and used to treat NPMSC resulted in NPMSC chondrogenesis	[131]
miR-140-3p (Detail, NA)	Its overexpression in NPMSC facilitated cell differentiation toward the NPC-like phenotype	[133]
Serum supplementation	ADMSCs cultured without FBS have enhanced potential for chondrogenic differentiation	[135]

ADMSC, adipose-derived MSC; AFC, annulus fibrosus cell; BMSC, bone marrow-derived MSCs; CDM, chondrogenic differentiation medium; Co-Q10, Coenzyme Q10; FBS, fetal bovine serum; miR, microRNA; Mkx, Mohawk; MSCs, mesenchymal stem cells; NA, not available; NPMSC, nucleus pulposus-derived MSC; NP, nucleus pulposus; SIRT1, silent mating type information regulator 2 homolog 1; SIX-1, sine oculis homeobox homolog-1; SOX9, SRY-Box Transcription Factor 9; TGF, transforming growth factor; Wnt, Wingless signaling transduction.

**Table 3 cells-12-02161-t003:** Exogenous factors for inducing mesenchymal stem cell differentiation into intervertebral disc cells.

Factors	Effects and Examples of Usage	References
O-vanillin (100 μM, Sigma-Aldrich, St. Louis, MO, USA)	The conditioned medium of o-vanillin-treated IVD cells induced the chondrogenic differentiation of MSCs	[138]
BSHXF (The First Hospital of Hunan University of Traditional Chinese Medicine).	ADMSCs exhibited differentiation toward an NPC-like phenotype using BSHXF-medicated serum	[140]
ASA VI (0.01–100 mg/L)	MSC differentiation into NP-like cells via regulating ERK1/2 and Smad2/3 signaling pathways	[141]
Salvianolic acid B (1–10 mg/L)	The chondrogenic differentiation of MSCs in vivo	[145]
PSO (Detail, NA)	Exerts various effects; ADMSC differentiation toward an NPC-like phenotype	[148]
Simvastatin (0.01–0.1 μM)	NPMSCs differentiate into NPC-like phenotypes following their treatments	[61]
PPS (5 μg/mL [163])	Potential to induce the chondrogenic differentiation of BMSCs	[163,164]

ADMSC, adipose-derived MSC; ASA VI, Asperosaponin VI; BMSC, bone marrow-derived MSCs; BSHXF, BuShenHuoXueFang; ERK, extracellular signal-regulated kinase; MSCs, mesenchymal stem cells; NA, not available; NPC, nucleus pulposus cell; NPMSC, nucleus pulposus-derived MSC; O-vanillin, Ortho-vanillin; PPS, pentosan polysulfate; PSO, psoralidin.

**Table 4 cells-12-02161-t004:** Conditioned mediums, exosomes, and co-cultures for inducing mesenchymal stem cell differentiation into intervertebral disc cells.

Factors	Effects and Examples of Usage	References
CM (NCCM, 1 × 10^6^ cells/5 mL [168]; NPC-CM, not specified, P3 [169])	NCCM exhibited a stronger effect on the chondrogenic differentiation of BMSCs compared to CDM in combination with hypoxia, which is necessary for NPC-CM to induce the chondrogenic differentiation of BMSCs	[168,169]
Exosomes (1 × 10^6^ cells/5 mL [170])	NPC exosomes promote the differentiation of BMSCs into an NPC-like phenotype mediated by the Notch 1 pathway	[170,171]
Co-culture (1 × 10^6^ cells/mL, 1:1 [56]; 2 × 10^6^ cells/mL, 1:1 [57]; 1 × 10^5^ cells/0.5 cm^3^, 1:1 [75]	The co-culture of MSCs with NPCs or AFCs differentiates MSCs into NPC or AFC-like cells; the co-culture of MSCs with IVD tissue differentiates MSCs into NPC-like cells in combination with dynamic compression or bilaminar cell pellets, which have synergistic effects	[56,57,68,72,75,76,177,178,179,180,181,182,183,184,185,186,187]

AFC, annulus fibrosus cell; BMSC, bone marrow-derived MSCs; CDM, chondrogenic differentiation medium; CM, conditioned medium; IVD, intervertebral disc; MSCs, mesenchymal stem cells; NCCM, conditioned medium of notochordal cells; NPC, nucleus pulposus cell; NPC-CM, NPC-conditioned medium.

**Table 5 cells-12-02161-t005:** Biomaterials for inducing mesenchymal stem cell differentiation into intervertebral disc cells.

Biomaterial		Differentiation	References
Hydrogel			
Natural	collagen #^1^	NP/chondrogenic	[189,190]
	gelatin #^2^		[191,192]
	HA		[193]
	alginate #^3^		[194]
	fibrin		[195]
	chitosan #^4^		[196,197,198]
	agarose		[199]
	polypeptide		[32,200]
	PRP and PRP/HA/batroxobin		[193,201]
	multiple materials combined		[202,203]
Synthetic	PEG	NP/chondrogenic	[204]
	polyacrylamide		[205]
	redox-polymerized carboxymethylcellulose		[206]
	methacrylated carboxymethylcellulose		[207]
	poly(N-isopropylacrylamide- N,N0-dimethylacrylamide-Laponite		[208]
	poly(acrylamide-co-acrylic acid) microhydrogels		[209]
	poly lactide-co-glycolide		[210,211]
	poly glycerol monomethacrylate-poly 2-hydroxypropyl methacrylate diblock copolymer		[58]
Hybrid	PPS incorporated PEG and HA	NP/chondrogenic	[212]
	a highly sulfated semi-synthetic polysaccharide combined with PEG/HA		[213]
	poly(N-isopropylacrylamide-graft-chondroitin sulfate) hydrogel combined with or without alginate microparticles		[98,214]
	poly D,L-lactide-co-glycolide nanoparticles carrying TGF-β3 in dextran/gelatin hydrogel		[215]
	1-ethyl-3(3-dimethyl aminopropyl) carbodiimide and N-hydroxysuccinimide cross-linked type II collagen/HA hydrogel		[203]
	nitrogen-doped plasma-polymerized ethylene		[216]
Commercial	Hydromatrix	NP/chondrogenic	[217]
	Puramatrix		
Other types of gels			
Natural	alginate	NP/chondrogenic	[56,57,220,221]
	poly L-lactic acid scaffolds		[118]
	collagen-based carriers		[219,222,223]
Synthetic	PEG diacrylate microcryogel		[224]
	a biocompatible KLD-12 polypeptide/TGF-β1 nanofiber gel		[225]
	layered double hydroxide nanoparticles		[226]
Other materials			
Decellularized ECM	simple decellularized NP-ECM; genipin-cross-linked decellularized NP hydrogel; genipin-cross-linked decellularized AF hydrogel; decellularized NP and AF ECM mixtures	NP/AF	[229,230,231,232,233]
Pellet culture	pellet culture of MSCs	NP/chondrogenic	[84]
Chimeric	a silk-based scaffold	NP/AF	[234,235]

AF, annulus fibrosus; HA, hyaluronic acid; NP, nucleus pulposus; PEG, polyethylene glycol; PPS, pentosan polysulfate; PRP, platelet-rich plasma; TGF, transforming growth factor; # indicates the order of potential for MSC differentiation into an NPC-like phenotype [227].

**Table 6 cells-12-02161-t006:** Environmental factors for inducing mesenchymal stem cell differentiation into intervertebral disc cells.

Factors	Effects and Examples of Usage	References
Hypoxia	Promotes MSC differentiation toward an NPC-like phenotype; confers agonistic effects to NPC-CM treatment on BMSC differentiation toward NPC-like cells; highly relevant to HIF-1α activity	[77,78,79,169,242,243,244]
Hyperosmolarity	Whether high osmotic pressure promotes the differentiation of MSCs into NPC-like cells remains controversial	[251,252,253,254]
Mechanical stresses	A low cyclic and dynamic compressive load and a simulated-physiological compressive load promote the differentiation of MSCs into NPC-like cells	[7,70,224,256,257,258,259,260,261,262,263,264]

BMSC, bone marrow-derived MSCs; CM, conditioned medium; HIF-1α, hypoxia-inducible factor-1α; MSC, mesenchymal stem cell; NPC, nucleus pulposus cell.

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
