# Peer review of "A Review: Methodologies to Promote the Differentiation of Mesenchymal Stem Cells for the Regeneration of Intervertebral Disc Cells Following Intervertebral Disc Degeneration"

_cells, 2023, doi:10.3390/cells12172161_

Round 1

Reviewer 1 Report

The article is well written and very clear. Research purposes, materials and methods are well explained. The topic of MSC therapy is highly impactful in the development of clinical therapies for YVD IDD. No changes are needed.

Author Response

Thank you for reviewing our paper and your constructive comments.

Reviewer 2 Report

Manuscript by Ohnishi et al., provides a comprehensive review of current status of MSCs as therapeutic agents for treatment of IVD degeneration (IDD). Authors have summarized and compared a variety of studies using different types of MSCs including BMSCs, ADPCs, UCMSCs and NP-MSCs for their potential to treat IDD in humans. Authors have also discussed about the possible mechanism of action and discussed about differentiation of MSCs in vitro using different growth factors. Review is well written, figures and tables are well organized and updated with references. In my opinion, authors could have strengthened the conclusion by comparing and contrasting some of these studies and provided future implications with emphasis on current limitations and its applications for clinical use.  

Author Response

Response: We appreciate your advice; consequently, we have modified the Conclusion section accordingly.

Line 554

  1. Conclusions -Highlights, limitations, and future perspectives

Line 558

Among the multiple types of growth factors, TGF-β superfamily members and GDF5 and 6 played the primary roles, with BMP-7 and GDF6 likely being the most effective to elicit MSC differentiation into NPC-like cells. However, the efficacies of other endogenous and exogenous molecular factors have not been studied. Moreover, their interactions remain unknown; for example, it is not known whether these molecules synergistically enhance each other’s’ effects, negate them, or have completely unrelated effects. This scarcity of information is likely the limitation for their clinical use. In contrast, biomaterials and environmental factors have been extensively studied in combination with these molecules, with most studies elucidating their synergistic effects. Hence, the combination of molecules, biomaterials, and environmental factors may be helpful in identifying a novel methodology for MSC differentiation. Numerous types of biomaterials—including natural, synthetic, and chimeric materials—have been studied; these materials simulate scaffolds of the ECM of IVD. Although synthetic and chimeric materials can be structurally sturdy, their application in clinical use is likely limited owing to the likelihood of foreign body reaction.

Reviewer 3 Report

Ohnishi et al. summarized the molecules, scaffolds, and environmental factors that facilitate the differentiation of MSCs into IVD cells for regenerative therapies of IDD. This is a cutting-edge topic with high importance. Minor concerns should be addressed before acceptance.

1. More latest literatures published in 2022 and 2023 should be concluded to increase the timeliness of this review.

2. In Section 3, more detailed introduction of different types of MSC with the relationship of IVD needs to be added.

3. Table 1-4 needs to be furnished with the physiological level and secretion source of different growth factors.

4. The performance of differentiation in different biomaterials should be summarized in Table 5.

5. The section of Future Perspectives need to be added to outlook the development of this area in the near future.

Reviewer 4 Report

 Suggestions for Innovation:

1.        It would be beneficial to emphasize any novel findings or approaches that have not been extensively covered in existing literature. Highlighting the specific advances or unique methodologies would enhance the article's innovation.\

2.        Consider including discussions on potential challenges and limitations related to the methodologies presented.  Identifying areas for improvement can stimulate further research and open new avenues for investigation.

Suggestions for content:

1.        In the introduction, elaborate on the current challenges in IVD treatment, such as the limited efficacy of available therapies and the need for regenerative treatments that address the underlying pathological conditions of IVD degeneration. Highlight the importance of tissue regeneration as an ideal treatment strategy for IDD.

2.        While discussing MSC therapy, emphasize the importance of selecting appropriate sources of MSCs for transplantation. Consider comparing the advantages and disadvantages of autologous versus allogeneic MSCs in the context of IVD regeneration, including factors like availability, immunogenicity, and potential complications. Here are some valuable references:

(1)        Liu, H.; Zhang, H.; Han, Y.; Hu, Y.; Geng, Z.; Su, J. Bacterial Extracellular Vesicles-Based Therapeutic Strategies for Bone and Soft Tissue Tumors Therapy. Theranostics 2022, 12 (15), 6576–6594. https://doi.org/10.7150/thno.78034.

(2)        Zhang, H.; Wu, S.; Chen, W.; Hu, Y.; Geng, Z.; Su, J. Bone/Cartilage Targeted Hydrogel: Strategies and Applications. Bioact. Mater. 2023, 23, 156–169. https://doi.org/10.1016/j.bioactmat.2022.10.028.

(3)        Wang, Y.; Zhang, H.; Hu, Y.; Jing, Y.; Geng, Z.; Su, J. Bone Repair Biomaterials: A Perspective from Immunomodulation. Adv. Funct. Mater. 2022, 32 (51), 2208639. https://doi.org/10.1002/adfm.202208639.

(4)        Abdalla, A. A.; Pendegrass, C. J. Biological Approaches to the Repair and Regeneration of the Rotator Cuff Tendon-Bone Enthesis: A Literature Review. Biomater. Transl. 2023, 4 (2), 85. https://doi.org/10.12336/biomatertransl.2023.02.004.

(5)        Peng, Y.; Qing, X.; Shu, H.; Tian, S.; Yang, W.; Chen, S.; Lin, H.; Lv, X.; Zhao, L.; Chen, X.; Pu, F.; Huang, D.; Cao, X.; Shao, Z. Proper Animal Experimental Designs for Preclinical Research of Biomaterials for Intervertebral Disc Regeneration. Biomater. Transl. 2021, 2 (2), 91. https://doi.org/10.12336/biomatertransl.2021.02.003.

3.        When discussing NPC-derived exosomes for differentiation of MSCs into NPC-like phenotypes, provide more information on the potential mechanisms of exosome-mediated differentiation. Elaborate on how these exosomes influence MSC behavior and promote their transformation into IVD cells.

4.        In the section on environmental factors, delve deeper into the specific harsh conditions present in the NP that may affect implanted MSC survival and biological behaviors. Provide insights into strategies that could mitigate the detrimental effects of the NP environment on MSCs, such as preconditioning or the use of protective scaffolds.

5.        Offer a critical analysis of the various scaffolds and environmental factors discussed for MSC differentiation into IVD cells. Compare their effectiveness and feasibility in clinical applications, considering factors like scalability, cost-effectiveness, and biocompatibility.

6.        Consider including a section that discusses the challenges and limitations of MSC-based regenerative therapies for IVD degeneration. Address issues like the potential risk of tumorigenicity, immune rejection, and long-term viability of implanted cells.

7.        When discussing the differentiation of MSCs into NPCs or AFCs, explore the potential differences in their regenerative capabilities and suitability for IVD repair. Highlight the specific roles these cell types play in the regeneration of different components of the IVD.

8.        In the discussion of molecules, scaffolds, and environmental factors, provide more specific details on the optimal combinations of these factors to promote MSC differentiation into functional IVD cells. Offer practical insights into how researchers and clinicians can apply this knowledge in regenerative treatments.

Overall, the English writing in the article is clear and well-structured, allowing readers to follow the presented information effectively.

1.        The introduction could be further refined by explicitly stating the review's main objective and its significance in the context of IDD and regenerative medicine.  Additionally, consider rephrasing certain sentences to enhance readability.

2.        Throughout the article, maintain consistent use of terminologies and abbreviations to avoid confusion.  Define any acronyms upon first use.

3.        Check for and correct any grammatical errors or awkward sentence structures that may distract readers from the main message.

4.        Ensure a smooth transition between sections and subsections, making it easier for readers to navigate through the article.
